# Investigating the Morphology, Optical, and Thermal Properties of Multiphase-TiO₂/MAPbI₃ Heterogeneous Thin-Films for Solar Cell Applications

**Thembinkosi D. Malevu** [1,*], **Tshwafo E. Motaung** [2,3], **Setumo V. Motloung** [1,4], **Lehlohonolo F. Koao** [5], **Teboho P. Mokoena** [1] and **Motlalepula R. Mhlongo** [1]

1 Department of Physics, Sefako Makgatho Health Sciences University, P.O. Box 94, Medunsa 0204, South Africa; cchataa@gmail.com (S.V.M.); teboho.mokoena@smu.ac.za (T.P.M.); rebecca.mhlongo@smu.ac.za (M.R.M.)
2 Department of Chemistry, Sefako Makgatho Health Sciences University, P.O. Box 235, Medunsa 0204, South Africa; motaungte@live.com
3 Department of Chemistry, School of Science, College of Science Engineering and Technology, University of South Africa, Preller Street, Muckleneuk Ridge, City of Tshwane, P.O. Box 392, Unisa 0003, South Africa
4 Department of Chemical and Physical Sciences, Walter Sisulu University, Private Bag X1, Mthatha 5117, South Africa
5 Department of Physics, University of the Free State (Qwa Qwa Campus), Private Bag X13, Phuthaditjhaba 9866, South Africa; koaolf@ufs.ac.za
* Correspondence: malevu.td@gmail.com

**Abstract:** The present study evaluates the effect of mesoporous multiphase titanium dioxide (TiO₂) nanoparticles (NPs) as an electron transporting layer and investigates the influence of phase composition on the perovskite solar cell (PSC) performances. This study also aims to evaluate PSC performance using conductive silver ink as an alternative counter electrode. The heterogeneous PSC thin-film solar cells were successfully fabricated and assembled by using a simple a doctor blade and two-step spin coating methods under ambient conditions. Scanning electron microscopy (SEM) micrograph images investigate methyl ammonium lead iodide (MAPbI₃) crystal formation on the mesoporous TiO₂ surface structure. Energy-dispersive x-ray spectroscopy (EDX) spectra reveal excellent qualitative and quantitative analysis corresponding to the SEM images in the TiO₂/MAPbI₃ heterogeneous thin films. Thermogravimetric analysis (TGA) characterization reveals that the TiO₂/MAPbI₃ thin films are thermally stable recording a maximum of 15.7% mass loss at 800 °C elevated temperatures. Photoluminescence spectroscopy (PL) characterized the effect of multiphase TiO₂ phase transformation on the TiO₂/MAPbI₃ recombination efficiencies. A maximum of 6% power conversion efficiency (PCE) with the open-circuit voltage (Voc) of $0.58 \pm 0.02$ V and short circuit current (Jsc) of $3.89 \pm 0.17 \, \mathrm{mAcm^{-2}}$ was achieved for devices with an active area of $3 \times 10^{-4} \, \mathrm{m^2}$ demonstrating that the synthesized multiphase TiO₂ nanoparticles are promising for large surface area manufacturing. Therefore, it is apparent that multiphase TiO₂ NPs play a significant role in the performance of the final device.

**Keywords:** multiphase TiO₂; mesoporous TiO₂; perovskite solar cells; doctor blade; conductive silver ink

## 1. Introduction

P- and n-type semiconducting transition metal oxides are extensively studied and put in use in third-generation solar cells ranging from dye-sensitized solar cells (DSSCs), quantum dot solar cells (QDSCs), organic solar cells (OSCs), and perovskite solar cells (PSCs) [1–3]. Thin films of TiO₂, ZnO, V₂O₅, MoO, CuO, and NiO₂ are used due to their nanostructural properties to facilitate charge extraction and transport

from photoactive hybrid materials [4–7]. The mesoporous structure in titanium dioxide ($TiO_2$) makes it a successful n-type photoanode electrode in photovoltaics (PVs). This unique $TiO_2$ morphology appears to serve as a host/absorber matrix for light active materials on its surface in DSSCs, QDSCs, and PSCs, which increases the metal oxide/light active material intermediate contact [8,9]. Tailoring the structure, morphology, and optical properties of metal oxides is key to optimizing the performance of solar cell devices [10–12]. Changes in $TiO_2$ merit properties such as phase composition, degree of crystallinity, porosity, crystalline size, particle size, and bandgap structure are significant to improving hybrid solar cell's photocurrent, open-circuit voltage, and energy conversion efficiencies.

Similar to DSSCs, $TiO_2$ thin films are coated over conductive glass substrates, annealed, and stained with a light sensitive $ABX_3$ perovskite material in PSCs. The annealing process is known to transform $TiO_2$ into a crystalline and conductive electron transport layer (ETL), remove residual impurities, precursor solvents, and encourage brookite, anatase, and rutile phases formation [13]. In addition, significant research has demonstrated that annealing alters $TiO_2$ merit properties, which subsequently influence the metal oxide/light active interface properties [13,14]. In PVs anatase, $TiO_2$ is the most preferred phase due to its higher open-circuit voltage, photoactivity, and surface area than rutile-based PV cells [15]. An important concept that has not received much attention is the use of mixed-phase (anatase: rutile) $TiO_2$ nanoparticles (NPs) in PSCs. Extensive research on mixed-phase $TiO_2$ photoanode has been carried out in DSSCs. The majority of these studies observe synergy at less than 20% rutile phase formation [16,17]. Material characterization results reveal that changes in phase composition bring about changes in $TiO_2$ merit properties such as crystallinity, morphology, and optical properties of the photoanode [18]. The authors [14] have previously reported similar changes and are included in the article to further demonstrate the influence of multiphase-$TiO_2$ ETL on PSCs performance. Existing studies on mixed-phase $TiO_2$ achieve phase composition through a mixture of two-phase different titanium precursors, (e.g., P25 and P90), the addition of phase controlling agents/additives, (e.g., $CH_3COOH$, EDTA, and $(NH_4)_2SO$), and annealing $TiO_2$ NPs [16,17,19–27]. Li et al. [17] fabricated mixed-phase $TiO_2$ photoelectrodes based on phase pure anatase and rutile $TiO_2$. The authors' results demonstrate that the addition of small rutile phase content into anatase enhances the photoelectrode absorption coefficient, improving the photocurrent density and efficiency in comparison to the phase pure PV cells. In addition, Yun et al. [16] prepared mixed-phase $TiO_2$ NPs by $(NH_4)_2SO_2$ hydrolysis. The concentration of the sulfate ion successfully tunes anatase and rutile phase composition. The authors' characterization results indicate that the improved photocurrent and efficiencies in comparison to pure phase devices are attributed to high light scattering by large rutile particles. However, large amounts of rutile phase content decrease surface area and as a consequence dramatically reduce the device's performance. In similar research work, Ruan et al. [19] supplement the explanation of synergy in mixed-phase $TiO_2$ by symbiosis in anatase and rutile surface morphologies, which facilitates maximum interparticle charge transfer.

The above synthesis of mixed-phase $TiO_2$ NPs results in two distinctive rod-like and spherical morphologies associated with rutile and anatase phases, respectively, with rutile morphology relative to anatase decreasing solar cell performances due to its low surface area and packing structure both phenomena decreasing the electron collection in the devices.

Earlier studies conducted by the authors Lekesi et al. [28] on the formation of multiphase titanium dioxide ($TiO_2$) suggested that increasing the annealing temperature of $TiO_2$ brings about changes in material merit properties such as porosity, particle size, crystallinity, and optical properties. Without any further treatment, the as-synthesized multiphase titanium dioxide ($TiO_2$) was employed in this study and used as is to fabricate heterogeneous thin films of $TiO_2/MAPbI_3$.

Therefore, the main aim of this work was to use the mesoporous multiphase-$TiO_2$ NPs to investigate the influence of phase composition on PSC performance, and additionally, to evaluate the PSC performance using conductive silver ink as an alternative counter electrode. The phase transformation is achieved through annealing $TiO_2$ NPs from 200 to 1200 °C. Scanning electron microscopy coupled with energy-dispersive x-ray spectroscopy (SEM-EDX) was used to investigate the perovskite $MAPbI_3$ crystal formation and interaction with $TiO_2$ ETL. Fourier transform infrared spectroscopy (FT-IR) was used to confirm functional groups within the multiphase-$TiO_2$/$MAPbI_3$ heterogeneous thin films. Thermogravimetric analysis (TGA) was used to determine the thermal behavior and confirm the existence of heterogeneous $TiO_2$ and $MAPbI_3$ mixture from the TGA degradations steps within a temperature range of 0 to 800 °C. Photoluminescence spectroscopy (PL) was used to study recombination efficiencies. The increasing $TiO_2$ particle size and rutile phase increase photoexcitation and efficient electron extraction from the $MAPbI_3$ light active material. A detailed explanation of the effect of annealing/phase composition on the PSC IV performance is given. Maximum efficiency of up to 6.7% is reported, attributed to factors discussed in SEM and PL.

## 2. Methods

*Perovskite Solar Cell (PSCs) Device Fabrication*

Indium-doped tin oxide (ITO) glass substrates (Sigma-Aldrich, Burlington, MA, USA) were first cleaned with a detergent and ultrasonicated three times using water, isopropanol, and ethanol for 20 min separately. The glass substrates were later allowed to dry at ambient atmosphere and thereafter subjected to Ultraviolet (UV)–Ozone cleaning for a period of 15 min. To form a dense compact $TiO_2$ blocking layer, the UV–Ozone-cleaned substrates were coated with a 0.15 M titanium diisopropoxide bis(acetylacetonate) (75% in isopropanol, Sigma-Aldrich) in 1-butanol by spin-coating at 2000 rpm for 20 s which were then heated at 100 °C for 5 min. After the coated glass substrates were allowed to cool at room temperature, the process was repeated two times using a 0.3 M titanium diisopropoxide bis(acetylacetonate). Following the three-times coating of titanium diisopropoxide bis(acetylacetonate) the glass substrates were then heated at 250 °C for 15 min and allowed to cool at room temperature. A few micrometers thick layer of mesoporous-$TiO_2$ photoelectrode was formed from a paste of the authors' previously synthesized $TiO_2$ [28]. The micrometer thick layers were coated over the compact layer using the doctor blade technique thereafter the substrates were heated at 250 °C for 45 min. From here the substrates/photoelectrodes were treated with a 0.02 M $TiCl_4$ (98%, Sigma-Aldrich) solution maintained at 80 °C for 10 min, washed with deionized water, and dried at 100 °C in an open atmosphere. A single halide perovskite layer of $CH_3NH_3PbI_3$-($MAPbI_3$) was formed by a two-step spin coating of $PbI_2$ and MAI at ambient conditions (relative humidity = 45–60%) to form $TiO_2$/$MAPbI_3$ heterogeneous thin films. Firstly, a solution of 0.02 mL of 1 M $PbI_2$ (4.61 g) in a solvent mixture of dimethyl formamide DMF: dimethyl sulfoxide DMSO (3:1) prepared at 70 °C was spin-coated over the mesoporous $TiO_2$ layers at 3000 rpm for 10 s without loading time. The ITO/c-$TiO_2$/m-$TiO_2$/$PbI_2$ glass substrates were then dried at 100 °C for 10 min and allowed to cool to room temperature. The perovskite layer was completed by spin-coating 0.2 mL of 0.038 M $CH_3NH_3I$ at 4000 rpm for 20 s with a loading time of 20 s. The ITO/c-$TiO_2$/m-$TiO_2$/$MAPbI_3$ glass substrates were then dried at 100 °C for 10 min. A 0.02 mL solution of spiro-MeOTAD was spin-coated over the perovskite bearing substrated at 3500 rpm for 35 s. The spiro-MeOTAD deposited solution was obtained by dissolving 0.072 g of spiro-MeOTAD (Sigma-Aldrich) in 1 mL chlorobenzene (Sigma-Aldrich) solvent. This was followed by the addition of 0.029 mL 4-*tert*-butyl pyridine (Sigma-Aldrich) and 0.018 mL (0.52 g) lithium bis(trifluoromethanesulfonyl)imide in 1 mL acetonitrile (Sigma-Aldrich) solutions. The counter electrode was obtained by using silver (Ag) ink on an opposite ITO substrate. Approximately 80% of the ITO substrate was covered with a deposition

mask and a conductive Ag ink (Sigma-Aldrich) was deposited over the exposed area via doctor blade technique. The solar cells were assembled by preheating the $TiO_2$ coated substrates at 100 °C for 5 min and thereafter clamped with the opposite Ag coated electrodes. The fabricated cells were then heated at 100 °C for 15 min to form solar cells with device architecture of ITO/c-$TiO_2$/m-$TiO_2$/MAPbI$_3$/Ag/ITO. A total of six (6) devices are reported by varying only the mesoporous $TiO_2$ electron transport layer and the remaining layers are kept constant. Figure 1 shows the fabrication steps of the different TiO2/MAPbI3 films.

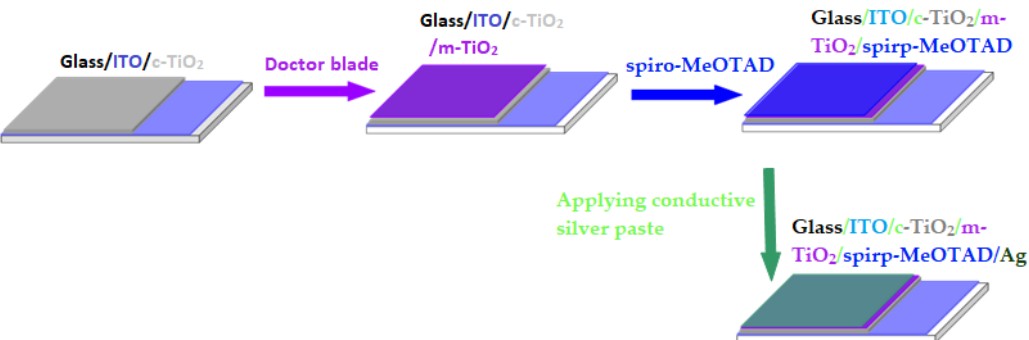

**Figure 1.** Schematic scheme representing steps involved in the fabricated device.

### 3. Characterization

To determine the surface morphology of $TiO_2$/MAPbI$_3$ heterogeneous thin-films, a TESCAN VEGAN3 scanning electron microscopy (OXFORD X-MAX$^N$) coupled with EDX operated at 20 kV was used and the analysis was done at room temperature. The surface of the samples was sputter-coated with carbon for 9.9 s to produce a conductive coating on the samples. To characterize the thermal behavior of the samples Perkin-Elmer STA6000 thermogravimetric analyzer (TGA) was used. A total of 20–25 mg of the photoanode/perovskite was obtained by scraping the $TiO_2$/MAPbI$_3$ heterogeneous thin films off the ITO conductive substrate. The samples were heated at 10 °C min$^{-1}$ from ambient to 800 °C under 20 mL min$^{-1}$ nitrogen flow. In this work, a Perkin-Elmer Spectrum 100 Fourier transform infrared (FT-IR) spectroscope fitted with an attenuated total reflection (ATR) detector was used. FT-IR spectrometer was used in this study to identify functional groups in the $TiO_2$/MAPbI$_3$ heterogeneous thin films. The infrared (IR) spectra were recorded between 4000 and 650 cm$^{-1}$ at a resolution of 4 cm$^{-1}$. Photoluminescence spectroscopy (PL) characterization was performed on as-prepared $TiO_2$ nanopowder, annealed $TiO_2$ nanopowder, and $TiO_2$/MAPbI$_3$ heterogeneous films under a Carry Eclipse Fluorescence spectrophotometer with a 150 W xenon lamp ($\lambda_{ex}$ = 238 nm). The electrical properties of the final devices with an active area of 3 cm$^2$ = 3 × 10$^{-4}$ m$^2$ were measured using Newport Oriel LCS-100 solar simulator and Keithley source meter (Keithley 2450) under AM1.5 G and 1000 W/m$^2$ standard irradiation test conditions calibrated from a standard Si-based solar cell.

### 4. Results and Discussion

*4.1. Morphology and Chemical Analysis*

Figure 2a–f shows low-magnification SEM surface morphology of the as-prep $TiO_2$/MAPbI$_3$ and annealed heterogeneous thin films, respectively. It should be mentioned that the low magnification was used in this work to allow us to have an overview of how the surface looks following the deposition of MAPbI$_3$ solution on top of multiphase $TiO_2$ nanomaterials. The surface images depict that annealing the multiphase-$TiO_2$ nanopowder has a strong influence on MAPbI$_3$ crystals formation over the $TiO_2$ surface structure. The images show two distinctive morphologies; (1) small particles (assumed to be nanometer-sized $TiO_2$ nanoparticles (NPs)) and (2) relatively large particles (assumed to be MAPbI$_3$ crystals). Similar perovskite crystal morphology is

reported by Christians et al. [22] from the two-step MAPbI$_3$ deposition method. In Figure 1 1t is observed that the increased annealing temperature in the multiphase-TiO$_2$ NPs encourages a non-uniform MAPbI$_3$ crystal dispersion increasing the distribution over the nanoporous TiO$_2$ surface. Im et al. [23] demonstrated comparable perovskite crystal growth. From all the SEM images the MAPbI$_3$ crystals are of the same size due to the consistent preparation method.

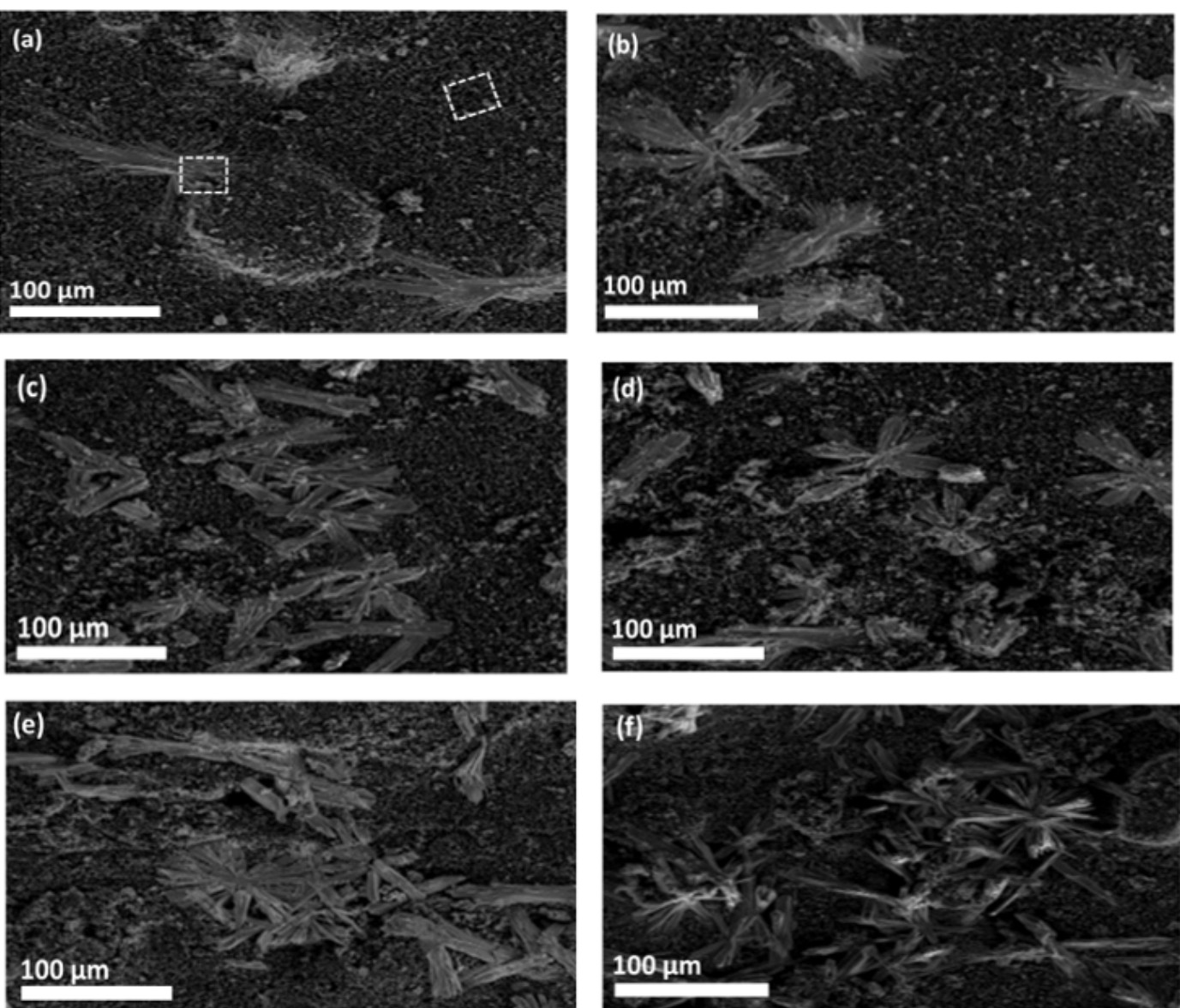

**Figure 2.** Low-magnification SEM micrographs presenting the surface morphology of the (**a**) as-prep TiO$_2$/MAPbI$_3$, (**b**) 200 °C/MAPbI$_3$ (**c**) 400 °C/MAPbI$_3$, (**d**) 800 °C/MAPbI$_3$, (**e**) 1000 °C/MAPbI$_3$, and (**f**) 1200 °C/MAPbI$_3$ heterogeneous thin-films.

Figure 3 shows high-magnification SEM micrographs and the corresponding SEM-EDX micrographs of TiO$_2$/MAPbI$_3$ heterogeneous thin-films used to investigate MAPbI$_3$ crystal formation onto the mesoporous TiO$_2$ NPs on a particular/selected region marked by white dotted squares in Figure 2a. The observed morphology from the magnified SEM image in Figure 3a shows an incomplete or initial stage of perovskite crystal formation. This is confirmed by the corresponding SEM-EDX micrograph presented in Figure 3b demonstrating the presence of C, Pb, and I perovskite main elements. It is anticipated in this study that the TiO$_2$ surface charging leads to possible perovskite nucleating (or

crystallization) sites depending on the MAPbI$_3$ surface charging density leading to the heterogeneous morphology shown in Figure 3c illustrating a cluster of crystals emerging from the surface of the TiO$_2$ nanoporous host material. Based on the observed MAPbI$_3$ crystal morphology, it is noted that there exists a preferred MAPbI$_3$ orientated crystal growth that propagates away from a fixed-point source justifying the proposed localized nucleating sites discussed above. Figure 3d demonstrates the presence of perovskite main elements similarly to Figure 3b.

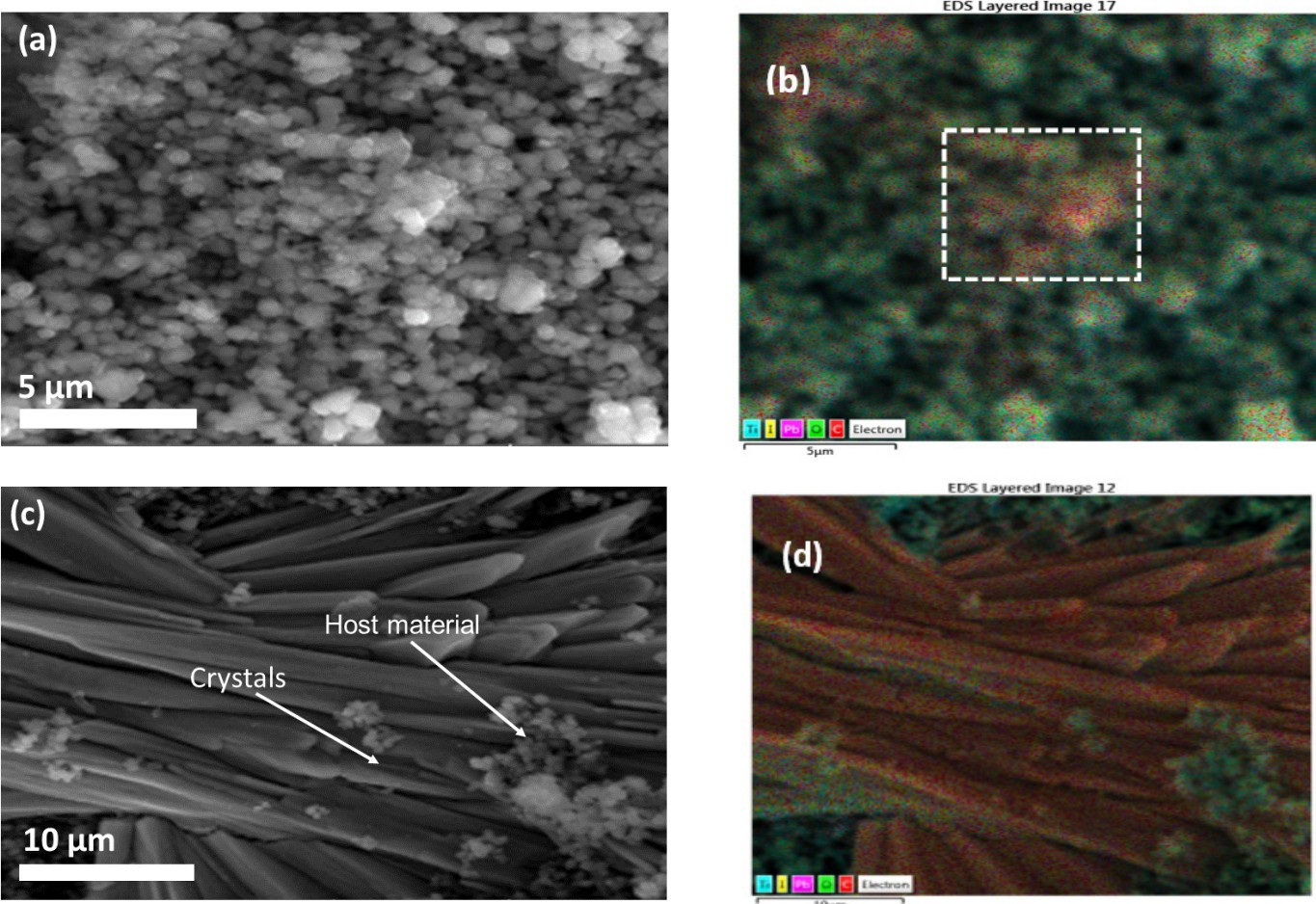

**Figure 3.** High-magnification SEM micrographs illustrating (**a**) surface morphology of as-prep TiO$_2$/MAPbI$_3$ interaction on ITO substrate, corresponding (**b**) SEM-EDX mapping image, (**c**) heterogeneous morphology interaction of MAPbI$_3$ nanocrystals embedded within TiO$_2$ nanopores, and corresponding (**d**) SEM-EDX image.

Figure 4a,b shows EDX spectra of as-prepared TiO$_2$/MAPbI$_3$, 200°C/MAPbI$_3$, and annealed MAPbI$_3$ heterogeneous thin films. In these results, all the expected elements such as Ti, O, Pb, and I were attained; however, the C element emerged from the carbon tape on which the samples were mounted. The EDX peak position (keV) and intensities (cps/eV) values provide excellent qualitative and quantitative analysis of the elements present in the TiO$_2$/MAPbI$_3$ heterogeneous thin films, respectively, consistent with SEM surface analysis. There were no significant differences worth mentioning in the proportion of expected elements in any of the annealed samples. The weight percent of Ti, O, Pb, and I elements obtained from EDX is nearly stoichiometric.

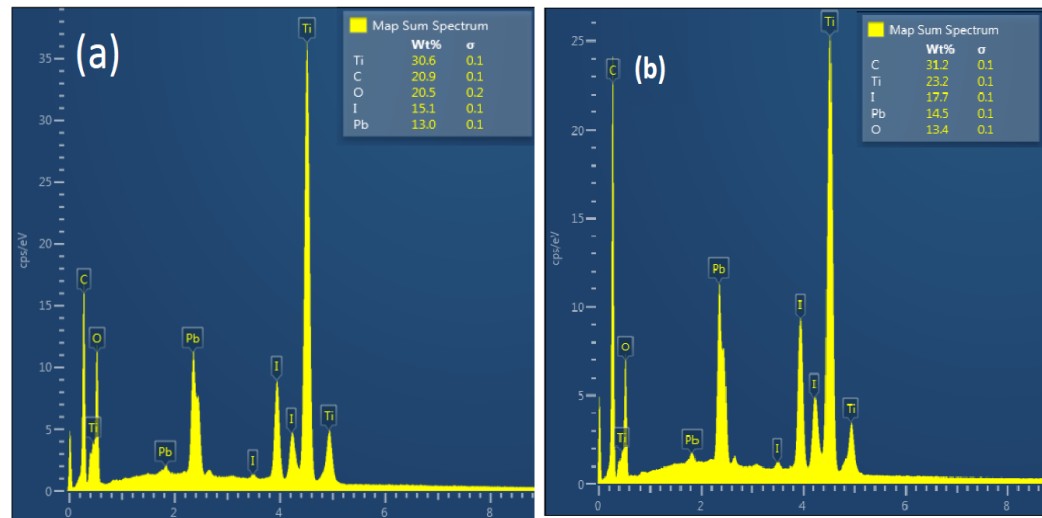

**Figure 4.** (**a,b**) Shows EDX spectra of as prepared $TiO_2/MAPbI_3$ and annealed 200 °C/$MAPbI_3$ heterogeneous thin films.

### 4.2. Fourier Transform Infrared Spectroscopy

FT-IR spectroscopy was used to investigate the compatibility between the $TiO_2$ NPs and $MAPbI_3$ crystal in the heterogeneous thin films. Figure 5 presents FT-IR spectra of the heterogeneous $TiO_2/MAPbI_3$ thin films. The activity of the IR-spectra was recorded between 4000 to 500 wavenumber regions. Vibrational modes from the heterogeneous thin films centered at 2902, 2804, 1703, and 709 $cm^{-1}$ were identified and correlated with SEM surface images. The large absorption bands located at low annealing temperatures at 2902, 2804, and 1703 $cm^{-1}$ are associated with low compatibility between the $TiO_2$ NPs and $MAPbI_3$ crystals. However, these vibrational bands systematically decrease with increasing $TiO_2$ annealing temperature, indicating that the interface contact compatibility in the heterogeneous thin films is improved. This can be successfully correlated with the increase in surface coverage from the SEM images. The appearance of a new vibrational mode located at 709 $cm^{-1}$ substantiates enhanced compatibility, indicating that a good interface contact has been established.

### 4.3. Thermogravimetric Analysis

To understand the thermal behavior and determine the degradation steps from the $TiO_2/MAPbI_3$ perovskite heterogeneous thin films TGA analysis was performed by heating the samples from ambient temperature to 800 °C in triplicate, average values are reported in Table 1. From the percentage mass loss in Figure 6a, it is apparent that all the heterogeneous thin films are thermally stable, recording a maximum mass loss percentage of 15.7% at high temperatures of up to 800 °C. The as-prep $TiO_2/MAPbI_3$ heterogeneous thin films experience the highest mass loss, recording a final residual mass percent of 84.3%. The remaining heterogeneous thin films increase systematically with increasing annealing temperature, exhibiting mass residues greater than 84.3%. A minimum of 2.2% mass loss is recorded for the 1200 °C/$MAPbI_3$ heterogeneous thin films. This shows that annealing of the $TiO_2$ nanopowder removes temperature-initiated degrading agents, resulting in thermally stable heterogeneous films. This observation is further depicted by the decreasing peak intensities in the DTGA graphs in Figure 6b. This systematic increase in the residual mass % in the heterogeneous films is further correlated to the $TiO_2$ phase transformation. $TiO_2$ NPs are naturally characterized by thermal transitions from the metastable phases to the stable rutile phase. Therefore, samples annealed at higher temperatures possess the thermodynamically stable rutile phase, whereas samples at lower annealing temperatures are metastable anatase phases. Hence, the 2.2% mass loss belongs to the heterogeneous samples bearing more stable-

TiO$_2$ NPs. In Figure 6a,b TiO$_2$/MAPbI$_3$ heterogeneous thin films decompose through three degradation mechanisms located at temperatures $\geq$ 106, 438, and 513 °C. The first and small degradation step located at ~106 °C is associated with the evaporation of absorbed water molecules on the surface of TiO$_2$ NPs, the second degradation step approximated at 438 °C is associated with the characteristic TiO$_2$ anatase to rutile phase transition [24,25]. As expected in both events, the mass loss % is significantly reduced with increasing annealing temperature as outlined above and summarized in Table 1. The third degradation step results from the decomposition of the MAPbI$_3$ perovskite by desorption of volatile MAI organic group [26].

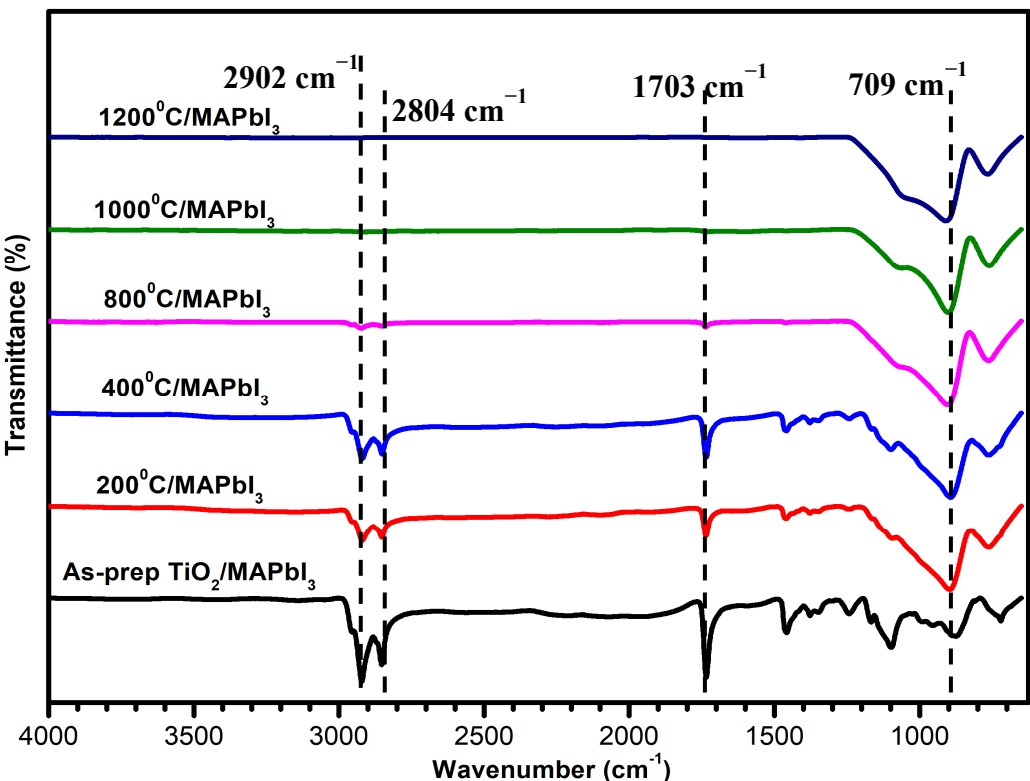

**Figure 5.** FT-IR spectra of the heterogeneous TiO$_2$/MAPbI$_3$ thin films.

**Table 1.** Presents DTGA peak temperatures, TGA degradation steps, and total mass loss from the TiO$_2$/MAPbI$_3$ heterogeneous thin films.

| Samples | Degradation Steps | | | Total Average Mass (%) Loss |
|---|---|---|---|---|
| | H$_2$O Evaporation | A-R Phase Transition | MAI Volatilization | |
| DTGA Temperatures (°C) | $\geq$106.2 | $\geq$437.5 | $\geq$512.5 | - |
| As-pre TiO$_2$/MAPbI$_3$ | 1.7% | 9.5% | 4.5% | 15.7% |
| 200/MAPbI$_3$ | 0.5% | 8.3% | 4.4% | 13.2% |
| 400/MAPbI$_3$ | 0% | 6.9% | 2.8% | 9.7% |
| 800/MAPbI$_3$ | 0% | 4.5% | 1.9% | 6.4% |
| 1000/MAPbI$_3$ | 0% | 3.5% | 1.2% | 4.7% |
| 1200/MAPbI$_3$ | 0% | 1.8% | 0.4% | 2.2% |

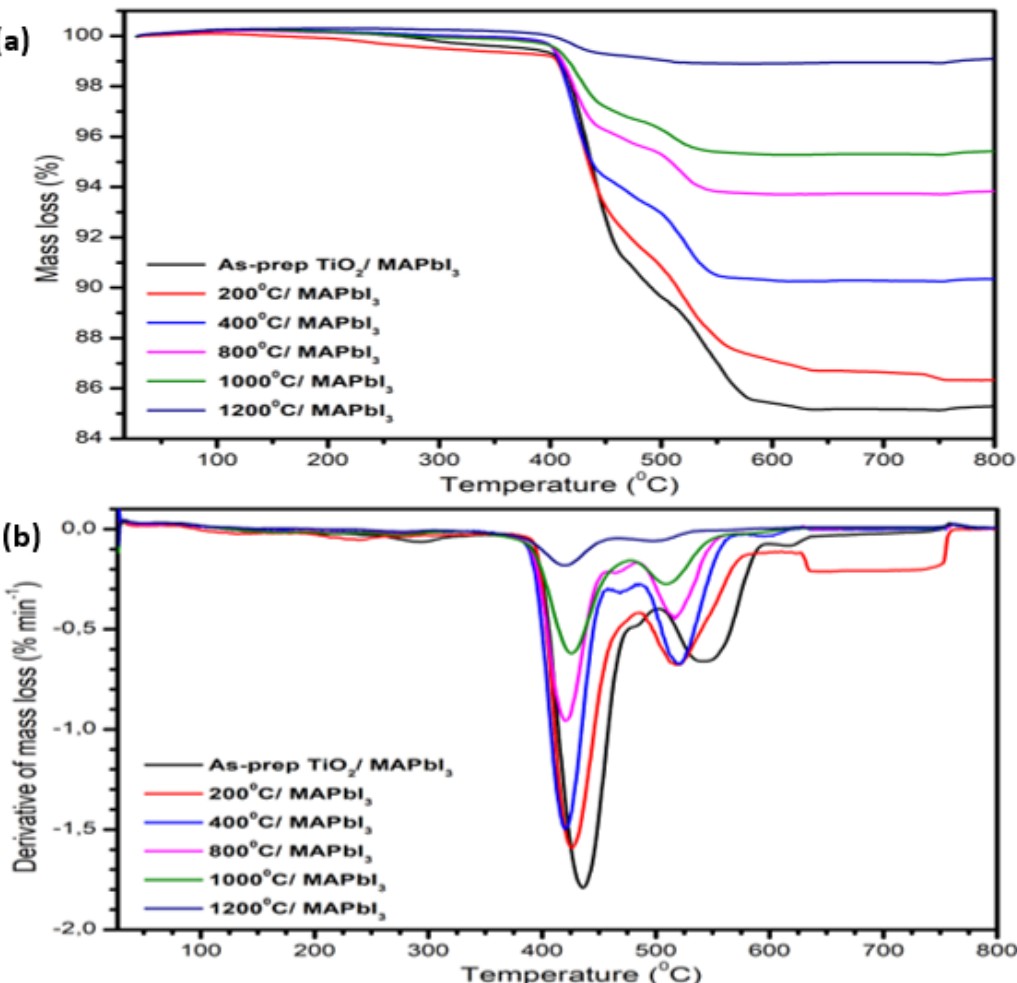

**Figure 6.** (**a**) TGA and (**b**) DTGA graphs present the effect of annealing $TiO_2$ NPs on the thermal degradation behavior of the resulting $TiO_2$/$MAPbI_3$ heterogeneous thin films.

### 4.4. Photoluminescence Spectroscopy

Photoluminescence (PL) is used to characterize photogenerated recombination pathways caused by defect levels within the bandgap structure of a semiconductor and the resulting PL peak emissions indicate the recombination of excitons, electrons, and holes between the conduction band and valence band. $TiO_2$ NPs are characterized by PL peak emissions caused by oxygen vacancies, surface states, and free and bound exciton recombinations [20,21,27]. The emission spectra in Figure 7a–c present PL peak emissions in the 300–700 nm range obtained from the excitation wavelength of $\lambda_{ex}$ = 238 nm for the as-prep $TiO_2$ nanopowder, annealed $TiO_2$ nanopowders, and as-prep $TiO_2$/$MAPbI_3$ and annealed heterogeneous thin films. The peak emissions were analyzed to investigate the effect of annealing/phase transformation on the recombination efficiencies. The deconvoluted gaussian fit for the as-prep $TiO_2$ nanopowder in Figure 7a shows characteristic peak emissions centered at $\lambda_{em}$ = 426, 466, 501, and 537 nm, a similar PL spectrum has previously been reported by Tsega et al. [27] and Acchutharaman et al. [21]. The prominent emissions located at 426 and 466 nm are associated with the oxygen vacancies and surface state dominant defect levels in the $TiO_2$ forbidden gap, whereas the shoulder peak emissions at high wavelengths of 501 and 537 nm are associated with the free edge and bound exciton minority recombinations. Figure 7b demonstrates the effect of annealing multiphase-$TiO_2$ on the material's recombination rates, insert of Figure 7b (normalized peak intensities) presents similar peak emissions reported in Figure 6a indicating that throughout the annealing process the peak emissions result from intrinsic trapping sites within the forbidden gap of

multiphase-TiO$_2$. From Figure 7b, it is observed that by increasing annealing temperature, the multiphase-TiO$_2$ NPs undergo PL quenching. The observed luminescence quenching may be attributed to several factors, including the presence of trap centers, grain boundary defects, and the band structure thermal quenching effect. In this study, the band structure thermal quenching is observed indicated by the decreasing trend in peak intensities of the samples as follows; from the samples: as-prep TiO$_2$ → 200 °C → 400 °C. This indicates that the intrinsic recombination rates (or concentration of defect levels) from oxygen vacancies, surface states, and excitons are significantly reduced. A similar observation has been reported by Fan et al. [20] in mixed-phase TiO$_2$ NPs and is ascribed to the increase in TiO$_2$ degree of crystallinity and the desorption of oxygen molecules reducing TiO$_2$ dominant defect levels. However, with a further increase in annealing temperature, a counteractive phenomenon emerges, substantially increasing the peak intensities dramatically. This apparent change in behavior from the PL peak emissions is attributed to the multiphase-TiO$_2$ phase transition and an increase in TiO$_2$ particle size. Comparable results are reported by Yun et al. [16] and Ruan et al. [19] demonstrating that more rutile phase formation and large particle size result in a low electron mobile phase and increased light-scattering, respectively, providing an efficient charge carrier recombination pathway.

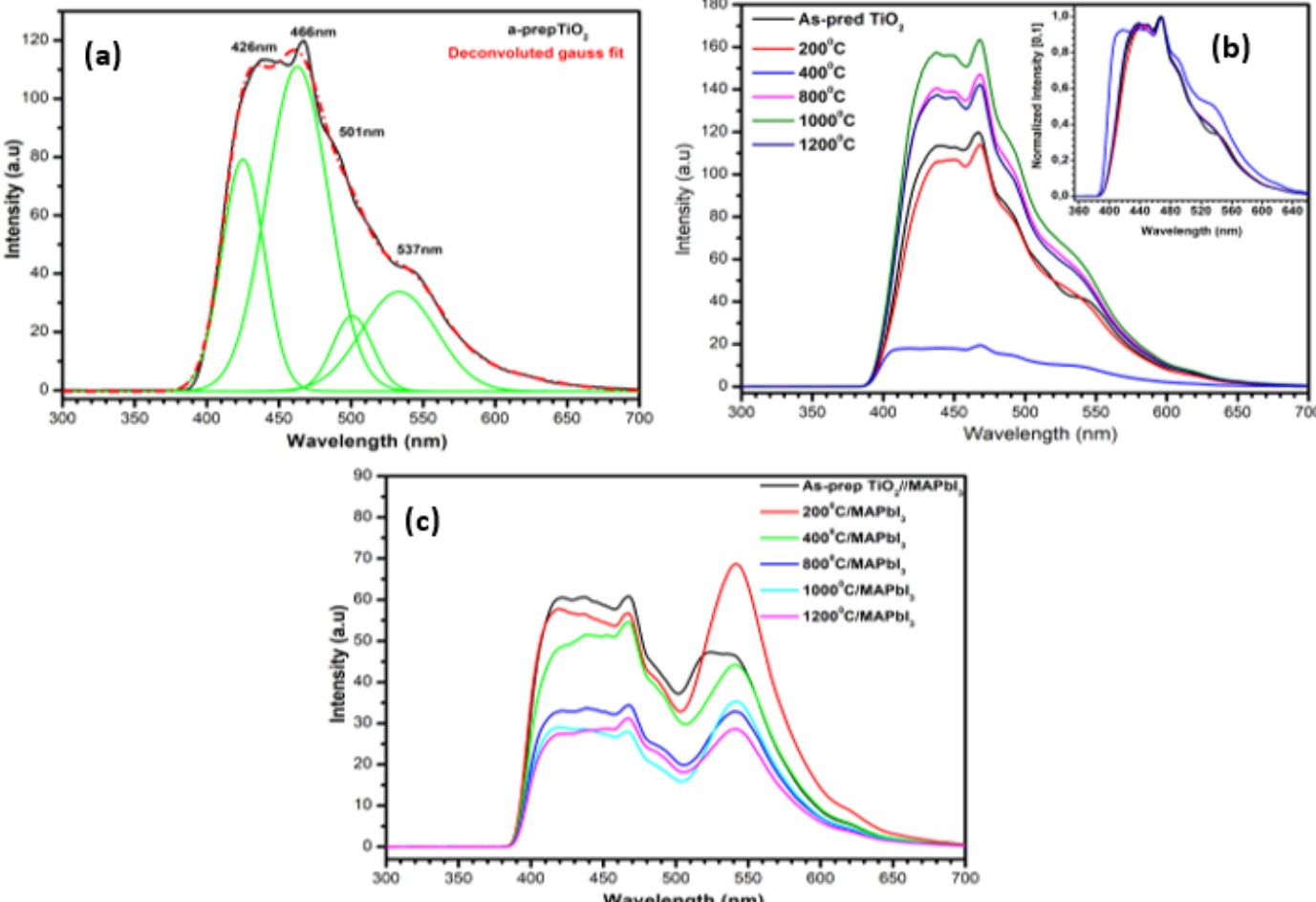

**Figure 7.** Presents PL peak emissions of the (**a**) deconvoluted as-prep TiO$_2$ nanopowder, (**b**) annealed TiO$_2$ nanopowders, and (**c**) as-prep TiO$_2$/MAPbI$_3$ and annealed heterogeneous thin films at $\lambda_{ex}$ = 238 nm excitation wavelength.

Figure 7c depicts PL peak emissions from the as-prep $TiO_2$/$MAPbI_3$ and annealed heterogeneous thin films. A new dominant peak emission in all the heterogeneous films located at 540 nm appears. This shows that an intermediate contact is established between the ETLs and the light active material; therefore, such recombination rate is attributed to $TiO_2$ ETL and $MAPbI_3$ interface electron decaying pathway. For comparison, it is observed that all the maximum peak intensities of the $TiO_2$/$MAPbI_3$ heterogeneous thin films are significantly reduced when compared to the maximum peak intensity of the as-prep $TiO_2$ nanopowder reported in Figure 7a,b. This behavior may indicate that the incorporation of the light active $MAPbI_3$ perovskite material encourages maximum electron collection as a result of minimum photon emissions. A systematic decrease in $TiO_2$/$MAPbI_3$ PL peak emission is observed with increasing annealing temperature/rutile phase content. This unusual trend highlights the synergistic effect of the multiphase-$TiO_2$ material influencing the overall photogenerated electron collection. The synergy results from the large photon scattering caused by the large particle size, which contributes to large photoexcitation in the perovskite material.

*4.5. Electrical Characterization*

Figure 8 depicts a simplified schematic image of the devices. Figure 9 shows IV characterization curves measured under AM1.5G and 1000 W/$m^2$ standard illumination conditions. Table 2 summarizes electrical average parameter values with standard deviations from the three best cells for each sample. The power conversion efficiency of the devices given in Table 2 is calculated from Equation (1). From the IV curves in Figure 9 and Table 2, it is observed that there is a prominent increase in current density ($J_{sc}$) and power conversion efficiency ($\eta$) from the PSCs with increasing rutile phase content and annealing temperature. This observation is correlated with the increasing $MAPbI_3$ surface coverage morphology (SEM) over the large multiphase-$TiO_2$ NPs, and the evident decrease in PL in Figure 7c, resulting in enhanced $J_{sc}$ and $\eta$. This indicates that the increase in the rutile phase establishes an efficient electron transfer from the photoexcited perovskite material into the conduction band of the anatase phase. A maximum of 6.7% efficiency is achieved for the solar cells fabricated from the 1000 °C/$MAPbI_3$ heterogeneous thin films. Therefore, the IV characterization demonstrates that although the rutile phase is characterized by slow electron mobility resulting in increased multiphase-$TiO_2$ recombination rates as shown in Figure 6b, the increasing PSC performance with increasing rutile content demonstrates that rutile is a high electron harvesting phase that extracts maximum photogenerated electrons from the perovskite material into the more electron mobile anatase phase decreasing the recombination as shown by Figure 7c. This is followed by a significant decrease to 5.1% efficiencies as rutile becomes the more dominant phase indicating that phase optimization in the multiphase-$TiO_2$ NPs is required for optimal PSC performances.

$$\eta = \frac{V_{oc} \cdot J_{sc} \cdot FF}{P_{in}} = \frac{V_{oc} \cdot J_{sc} \cdot FF}{A \cdot 1000 \text{ W}/m} \tag{1}$$

Equation (1) is used in the calculation of the power conversion efficiency ($\eta$), whereby $V_{oc}$ represents open-circuit voltage, $J_{sc}$ represents short circuit current density, FF represents the fill factor, A represents the active area of the solar cells, and $P_{in}$ represents the maximum power from the solar simulator lamp.

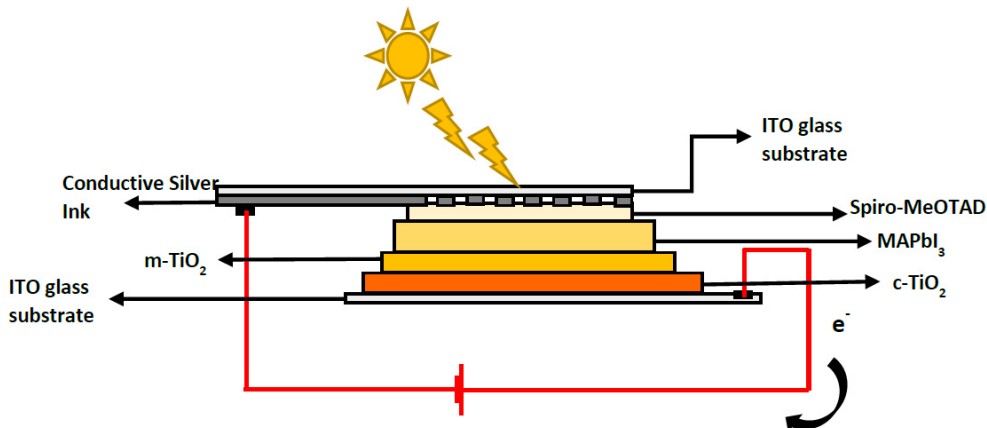

**Figure 8.** Schematic representation of the fabricated solar cell with device architecture ITO/c-TiO$_2$/m-TiO$_2$/MAPbI$_3$/Spiro-MeoTAD/Conductive Ag ink/ITO.

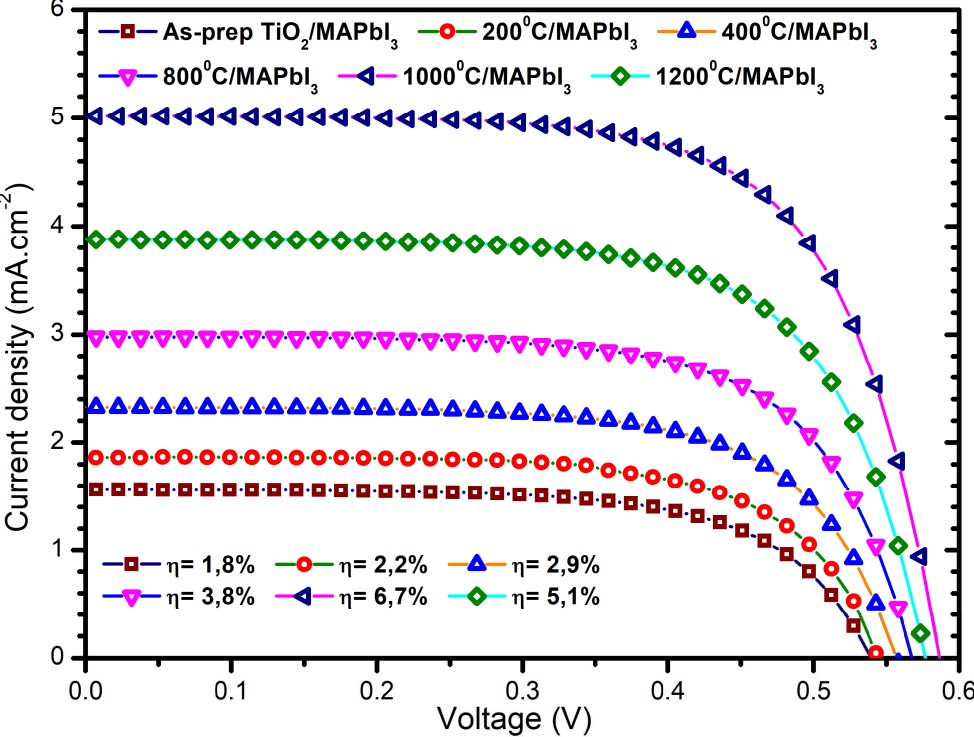

**Figure 9.** IV characterization curves.

**Table 2.** Photovoltaic parameters for the PSCs with device architecture ITO/c-TiO$_2$/m-TiO$_2$/MAPbI$_3$/Spiro-MeoTAD/Conductive Ag ink/ITO.

| Device | Phase Composition (A/R) | $V_{oc}$ (V) | $J_{sc}$ (mA·cm$^{-2}$) | FF (%) | $\eta$ (%) |
|---|---|---|---|---|---|
| As-pre TiO$_2$/MAPbI$_3$ | 68/32 | 0.53 ± 0.02 | 1.57 ± 0.15 | 65.7 ± 0.028 | 1.82 ± 0.55 |
| 200/MAPbI$_3$ | 65/35 | 0.54 ± 0.02 | 1.86 ± 0.13 | 66.5 ± 0.015 | 2.22 ± 0.42 |
| 400/MAPbI$_3$ | 46/54 | 0.56 ± 0.02 | 2.31 ± 0.11 | 66.8 ± 0.023 | 2.88 ± 0.58 |
| 800/MAPbI$_3$ | 60/40 | 0.57 ± 0.02 | 2.98 ± 0.18 | 67.4 ± 0.021 | 3.82 ± 0.72 |
| 1000/MAPbI$_3$ | 36/64 | 0.59 ± 0.02 | 5.02 ± 0.16 | 68.1 ± 0.025 | 6.72 ± 0.68 |
| 1200/MAPbI$_3$ | 34/66 | 0.58 ± 0.02 | 3.89 ± 0.17 | 68.0 ± 0.018 | 5.11 ± 0.64 |

## 5. Conclusions

In this study, the use of multiphase $TiO_2$ nanoparticles proved to be an excellent mediator for maximum electron collection of the light active $MAPbI_3$ perovskite material to the ITO glass substrate. Heterogeneous thin films of $TiO_2$ and $MAPbI_3$ for solar cells were successfully fabricated from doctor blade and two-step spin coating methods. In this work, the silver conductive ink was used as a counter electrode. The SEM-EDX micrographs reveal that the $MAPbI_3$ crystal formation mechanism involved $TiO_2$ pore filling (nucleation) followed by crystallization (growth) during the evaporation of the deposition solvents. The thermal analysis reveals that the $TiO_2$/$MAPbI_3$ thin films are thermally stable, recording a maximum of 15.7% mass loss at elevated temperatures. A systematic decrease in $TiO_2$/$MAPbI_3$ PL peak emission is observed with increasing annealing temperature/rutile phase content. This unusual trend highlights synergy from the multiphase-$TiO_2$ material influencing the overall photogenerated electron collection. Finally, PSCs of device structure ITO/c-$TiO_2$/m-$TiO_2$/$MAPbI_3$/Spiro-MeoTAD/Conductive Ag ink/ITO were fabricated and their performance was evaluated using the Keithley solar simulator. A maximum of 6.7% efficiency is achieved for the solar cells fabricated from the 1000 °C/$MAPbI_3$ heterogeneous thin films.

**Author Contributions:** The authors mentioned in this manuscript have participated equally in the content, design, analysis, writing and revising the manuscript. Also, each author certifies that this material has not been submitted to other journal for consideration or published in any other publication. All authors have read and agreed to the published version of the manuscript.

**Funding:** This work is funded by NRF Thuthuka SA, Grant No 137775.

**Institutional Review Board Statement:** Not applicable.

**Informed Consent Statement:** Not applicable.

**Data Availability Statement:** The data is available upon request.

**Acknowledgments:** I acknowledge the South African National Research Foundation through the Thuthuka programme (Grant No 137775) for the financial support throughout the course of the study. This work was not going to be successful without the financial assistance provided. This work is also dedicated to the late student Lehlohonolo Lekesi.

**Conflicts of Interest:** The authors declare that there is no conflict of interest.

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
