# Peer review of "Investigating the Morphology, Optical, and Thermal Properties of Multiphase-TiO2/MAPbI3 Heterogeneous Thin-Films for Solar Cell Applications"

_condensedmatter, doi:10.3390/condmat7020039_

Round 1

Reviewer 1 Report

 In light of prior work on perovskite solar cell for concentrated photovoltaics, including your own, I regret that we are unable to conclude that the paper provides the sort of significant conceptual advance in scientific understanding that would be of immediate interest to a broad readership of researchers in the energy community. I therefore feel that the present manuscript would be better suited to a more specialised journal than Condensed Matter. 

Author Response

The author thanks all the Reviewers for their valuable comments and suggestions. Below, the author’s responses are in red.

Reviewer 2 Report

The work was well-designed and the manuscript was well-written. It has a maximum of 6% power conversion efficiency (PCE) with the open circuit voltage (Voc) of about 0.58 V, which is promising for a solar cell. However, some points need to be addressed to improve the overall paper quality as follows.

Device assignment was not consistent, some with additional degree symbol (o) and the other not (Table 2).

Table 1 does not seem to be needed, because the device names have  been embedded in other figures and  tables.

Fig 3 would be much clearer if has white background.

Fig 4 may be improved for clarity. 

Fig 7 needs to improve, it is cluttered.

Author Response

(The authors gave the same response as above.)

Reviewer 3 Report

The manuscript reports the morphology, optical and thermal properties of multiphase-TiO2/MAPbI3 heterogeneous thin-films for solar cell applications. The authors used the mesoporous multiphase-TiO2 NPs to investigate the influence of phase composition on PSC performance. The manuscript should be published in Condensed Matter after authors amend some minor items listed below.

1. It is necessary to highlight the value of the experimental results to the application of solar cells. For example,  improving the efficiency of energy conversion or reducing manufacturing costs.

2. The sentence "Figure 3. shows EDX spectra of as prepared TiO2/MAPbI3 (a), 200℃/MAPbI3 (b), 400℃/MAPbI3 (c),
800℃/MAPbI3 (b), 1000℃/MAPbI3 (b), and 1200℃/MAPbI3 heterogeneous thin films. " in Page 6, should be "Figure 3. shows EDX spectra of as prepared TiO2/MAPbI3 (a), 200℃/MAPbI3 (d), 400℃/MAPbI3 (c), 800℃/MAPbI3 (e), 1000℃/MAPbI3 (b), and 1200℃/MAPbI3 heterogeneous thin films. "

3. Please analyze the cause of the change in the proportion of elements between the different pictures in Figure 3.

Author Response

(The authors gave the same response as above.)

Reviewer 4 Report

Malevu et al. report the fabrication and characterization of multiphase-TiO2/MAPbI3 thin films for solar cells. This work, despite of following a routine way and low efficiencies, might be publishable. However, the manuscript should be significantly revised and the following points should be addressed.

  1. Please elaborate the rationales behind the multiphase benefits. In particular, what kind of phase corresponds to the optimal one at the atomic level? Why? More mechanistic view should be provided!

  1. A figure illustrating the fabrication steps of the different TiO2/MAPbI3 films should be provided.

  1. Figure 3. Please improve the quality. It looks like raw data rather than satisfying the publication standard.

Figure 1 and Figure 2a: please include the size bar.

  1. English and format should be significantly improved:

Also, to evaluate the PSC performance using conductive silver ink as an alternative counter electrode.

No keywords

Introduction: there is a strange empty space between “Earlier studies…”

Figure 7: what is “m-“?

Author Response

(The authors gave the same response as above.)

Round 2

Reviewer 1 Report

The authors have addressed all the comments and I find it is suitable for consideration for publication in the journal.

Reviewer 4 Report

Accept